Corrected: Author correction; Author correction

# Metal-coordinated sub-10 nm membranes for water purification

Xinda You [1,2], Hong Wu[1,2,3], Runnan Zhang[1,2], Yanlei Su[1,2], Li Cao[1,2], Qianqian Yu[1,2], Jinqiu Yuan[1,2], Ke Xiao[1,2], Mingrui He[1,2] & Zhongyi Jiang[1,2]

Ultrathin membranes with potentially high permeability are urgently demanded in water purification. However, their facile, controllable fabrication remains a grand challenge. Herein, we demonstrate a metal-coordinated approach towards defect-free and robust membranes with sub-10 nm thickness. Phytic acid, a natural strong electron donor, is assembled with metal ion-based electron acceptors to fabricate metal-organophosphate membranes (MOPMs) in aqueous solution. Metal ions with higher binding energy or ionization potential such as $Fe^{3+}$ and $Zr^{4+}$ can generate defect-free structure while MOPM-$Fe^{3+}$ with super-hydrophilicity is preferred. The membrane thickness is minimized to 8 nm by varying the ligand concentration and the pore structure of MOPM-$Fe^{3+}$ is regulated by varying the $Fe^{3+}$ content. The membrane with optimized MOPM-$Fe^{3+}$ composition exhibits prominent water permeance ($109.8\,L\,m^{-2}\,h^{-1}\,bar^{-1}$) with dye rejections above 95% and superior stability. This strong-coordination assembly may enlighten the development of ultrathin high-performance membranes.

[1] Key Laboratory for Green Chemical Technology, School of Chemical Engineering and Technology, Tianjin University, Tianjin 300072, China. [2] Collaborative Innovation Center of Chemical Science and Engineering (Tianjin), Tianjin 300072, China. [3] Tianjin Key Laboratory of Membrane Science and Desalination Technology, Tianjin University, Tianjin 300072, China. Correspondence and requests for materials should be addressed to H.W. (email: wuhong@tju.edu.cn) or to Z.J. (email: zhyjiang@tju.edu.cn)

Membrane-based technology is emerging as a promising energy-efficient and environment-benign essential technology in water purification[1,2]. Since membrane permeation is inversely proportional to thickness, engineering ultrathin membranes with potentially high permeability and sufficient selectivity is critical to maximize membrane productivity and reduce membrane area[3]. As the basic building unit of living organisms, cell membrane sets a benchmark owing to the 7−8 nm thickness and exceptional permselectivity[4]. However, fabrication of sub-10 nm synthetic membranes which can simultaneously realize structural integrity and stability remains a grand challenge[5].

The interactions among membrane building units are the pivotal factors to create ever-thinner membranes which usually function as selective layers on porous substrates to form thin-film-composite (TFC) architecture for diverse applications. Interfacial polymerization (IP) is a technologically mature approach to prepare commercial TFC membranes as represented by the broadly utilized polyamide-based reverse osmosis and nanofiltration membranes[2]. Based on strong covalent bonds (hundreds of kJ mol$^{-1}$)[6], the IP process features extremely fast reaction between biphasic monomers at aqueous-organic interface and generates a stable polyamide layer spanning tens to hundreds of nanometers thick in seconds[7]. Hitherto, extensive efforts have been devoted towards control of IP process via substrate surface engineering[5], interfacial morphology regulating[8,9] and burgeoning 3D printing[10] for sub-100 nm polyamide membranes but with rare reports in obtaining sub-10 nm membranes because of the difficulty in precisely controlling the formation rate of covalent bonds[3,11]. Self-assembly using weak noncovalent interactions (10−30 kJ mol$^{-1}$)[6] with preferable controllability provides an alternative to prepare ultrathin membranes under mild conditions and enables precisely-tuned membrane thickness at nanometer resolution[2,12]. Polyelectrolytes ($M_w$ of $10^4$−$10^6$ Da) with either negative charge or positive charge are the currently popular units for electrostatic assembly, but since it is difficult to achieve the desired membrane structure in one assembly cycle[13], a large number of cycles (20−80 bilayers) and tens-of-nanometers thickness are normally required[14]. Additionally, polyelectrolytes-assembled membranes inevitably suffer from structural fluctuation or even disassemble in aqueous environment because of the strong interaction between water/solute and membrane-forming moieties[15]. Consequently, chemical crosslinking becomes the indispensable step[16]. The emerging two-dimensional assembly units represented by graphene oxide (GO) nanosheets could be readily assembled/stacked into paper-like membranes with a thickness around 20 nm via π–π stacking interactions (~10 kJ mol$^{-1}$)[17,18]. GO and its derivatives have been recognized as a new-generation material platform for ultrathin membranes with intrinsic fast water transport channels[19], whereas their aqueous instability remains a tricky issue due to the weak interaction[2]. Developing an assembly process based on moderate molecular interaction may offer an innovative strategy to fabricate membranes featuring ultrathin thickness, defect-free structure, and long-term superior stability.

Metal-organic coordination, an unconventional covalent bond interaction with tunable and moderate-intensity between that of strong covalent bond and that of weak noncovalent bond, affords a versatile toolbox for diversiform nanostructures since the available metal ions comprise almost half of the periodic table[20]. Notably, the ligand type significantly influences the coordination characters and needs to be rationally designed on demand. Metal-phenolic coordination evolves a powerful platform chemistry for ultrathin coating[21] or film[22] enlightened by water-stable $Fe^{3+}$-tannin (TA) films and capsules with variable thickness of 8−12 nm[23]. The weakly electron-donating phenolic ligands enable the smart disassembly of the metal-polyphenolic network in acid condition, which have been extensively utilized in drug delivery[22,23], but obviously not suitable for separation membrane as far as high stability is concerned. Compared with the metal-phenolic coordination, organophosphate ligands with much stronger electron donor nature can generate robust metal-organic bonds bearing puissant hydrolysis stability, and their abundant oxygen atoms capable of coordination offer structural variety[24,25]. The high coordination intensity between metal ion and organophosphate ligand has been demonstrated in genetic engineering where the cleaving process of sturdy phosphodiesters in DNA with a half-life of 130,000 years (25 °C) can be mediated by this metal-organophosphate interaction[26,27].

Herein, we propose a strong-coordination-based assembly strategy to fabricate ultrathin metal-organophosphate membranes (MOPMs) under aqueous condition. Phytic acid (PA) is a widely existing natural organic polyphosphate, bearing six phosphate groups and playing critical roles in biochemical processes of grains and seeds[28,29]. The multi-dentate character is expected to render the exceptional accessibility to form metal-organic networks and the strong metal-organophosphate interaction to generate stable membrane structure distinctive from conventional metal-polyphenolic networks. By altering the coordination metal ions with different ionization potential, tunable metal-organic interactions can be achieved to modulate physicochemical characteristics of MOPMs as demonstrated by both simulation and experimental results. For high water permeance and high solute rejection, the membrane thickness is minimized to 8 nm by reducing the ligand concentration and the pore structure of MOPMs is regulated by varying the metal ion content. This work demonstrates that the metal-organophosphate interaction could offer a moderate-interaction-based strategy to produce ultrathin membranes with high permeability, operation stability, and rich structure diversity.

## Results

**Metal-coordinated assembly for MOPMs**. In-situ fabrication of separation membranes on porous substrates to form ultrathin, robust structure is of practical significance. In this work, we assembled ultrathin membranes directly on ultrafiltration substrate. As shown in Fig. 1a, the partially-hydrolyzed poly-acrylonitrile (PAN) substrate was immersed upside-down in PA aqueous solution to form self-assembled molecular layer via H-bond interaction between the carboxyl groups on hydrolyzed PAN and the phosphate groups on PA (Supplementary Fig. 1). After that, the solution containing transition metal ions was added to trigger the coordination assembly, during which the deprotonated hydroxyl oxygen of phosphate group conveyed lone pair electrons to transition metal ion and formed coordination bonds. Thereby, the metal ions jointed the PA molecules together to generate transparent MOPMs followed by thermal curing at 60 °C. According to the soft-hard acid-base theory[30], the metal-organic coordination is influenced by polarization of ligand-donor atoms (Lewis base) and metal ions (Lewis acid), where species prefer to react with partners of the same type, i.e. hard ligands prefer hard metal ions whereas soft ligands prefer soft metal ions. Hence, we altered transition metal ions referring to soft acid, borderline acid, and hard acid, respectively, to manipulate the physicochemical structure of resultant MOPMs. The electron acceptability (polarizing power) of various metal ions is quantified by ionization potential ($I_n$) in the order of $Ag^+ < Zn^{2+} < Ni^{2+} < Fe^{3+} < Zr^{4+}$, reflecting the hardness of Lewis acid (Fig. 1b and Supplementary Table 1).

We firstly constructed a binary system composed of methyl phosphate and metal ion to predict the coordination intensity by

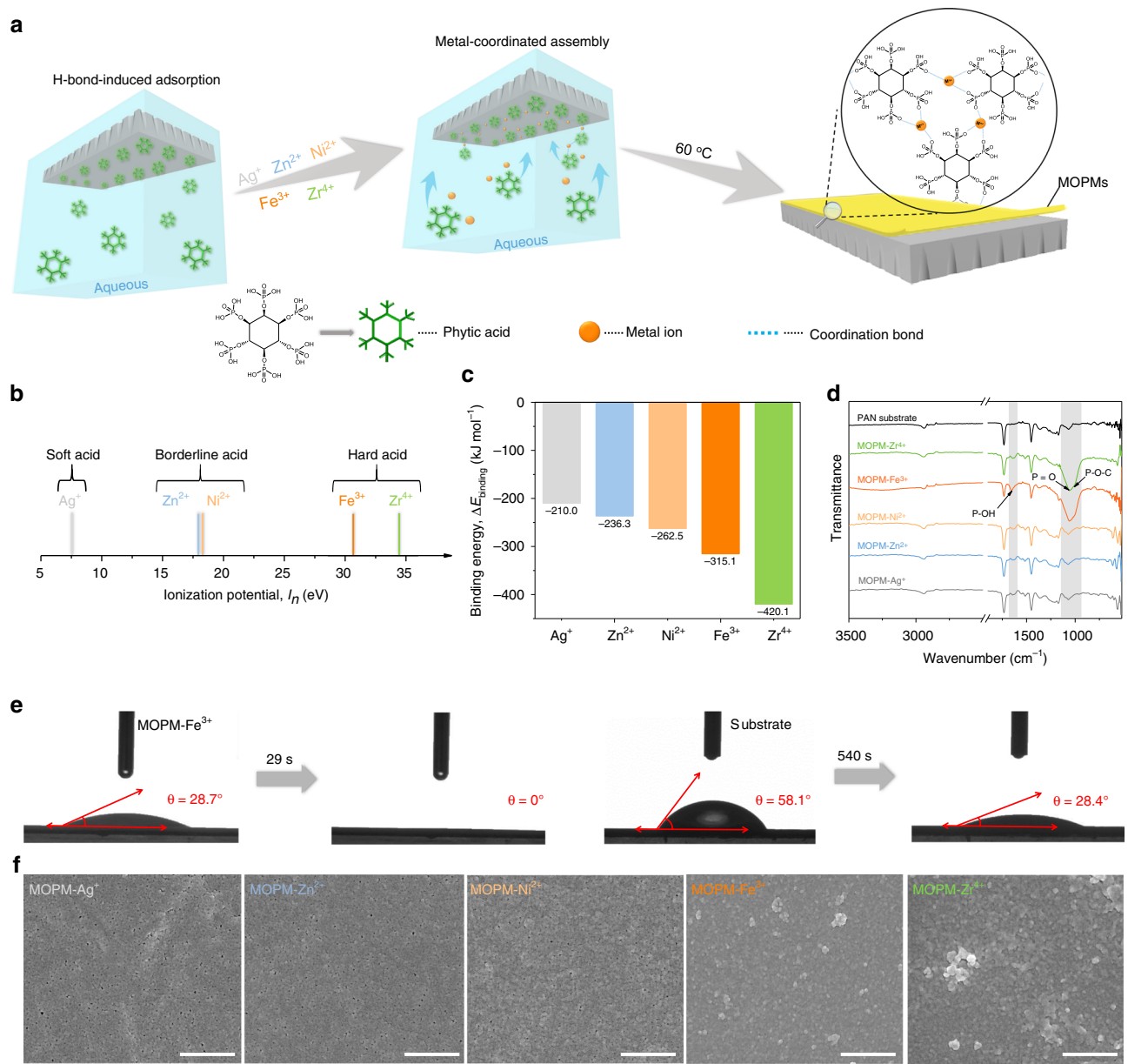

**Fig. 1** Metal-coordinated assembly of MOPMs. **a** Schematic illustration of the assembly process for MOPMs on PAN substrate with $Ag^+$, $Zn^{2+}$, $Ni^{2+}$, $Fe^{3+}$, and $Zr^{4+}$ ions. **b** Transition metal ions arranged based on ionization potential ($I_n$, eV) value. The ionization potential defined as: $I_n = E(M) - E(M^{n+})$, where $n = 1, 2, 3, 4$. **c** Calculated binding energy ($\Delta E_{binding}$, kJ mol$^{-1}$) between transition metal ions and deprotonated methyl phosphate. **d** FTIR curves of PAN substrate and MOPMs. **e** Digital photo images of water droplets on PAN substrate and MOPM-$Fe^{3+}$ membrane. **f** SEM images of MOPMs. Scale bar: 500 nm

molecular dynamics simulation (Supplementary Fig. 2). As shown in Fig. 1c, along with the increased ionization potential, the binding energy of metal-organophosphate coordination varies from $-210.0$ kJ mol$^{-1}$ for $Ag^+$ (monovalent ion, soft acid), $-236.3$ kJ mol$^{-1}$ for $Zn^{2+}$ and $-262.5$ kJ mol$^{-1}$ for $Ni^{2+}$ (divalent ions, borderline acid) to larger negative values of $-315.1$ kJ mol$^{-1}$ for $Fe^{3+}$ (trivalent ion, hard acid) and $-420.1$ kJ mol$^{-1}$ for $Zr^{4+}$ (tetravalent ion, hard acid), indicating the strongly-coordinated property approaching the intensity of covalent bonds. The increased polarizing power of metal ions intensifies the deformation of the electron cloud, thus enhancing the covalent feature of metal-organic bonds. The membrane-forming properties of PA with the abovementioned coordination metal ions were further investigated.

Successful formation of MOPMs on PAN substrate is confirmed by the metal and phosphorus elemental mappings (Supplementary Fig. 2). The Fourier transform infrared (FTIR) spectra of all the MOPMs display characteristic P–O–H, P=O and P–O–C bands at 1656 cm$^{-1}$, 1056 cm$^{-1}$ and 1004 cm$^{-1}$, respectively (Fig. 1d), revealing the existence of PA. The more intensive characteristic bands from PA in the spectra of MOPM-$Fe^{3+}$ and MOPM-$Zr^{4+}$ than that of others suggests the higher PA content. Since the P–O–H groups with a considerable hydration energy of 44.4 kJ mol$^{-1}$ are able to adsorb water molecules and render MOPMs with water affinity, the MOPM-$Fe^{3+}$ with the most intensive P–O–H band exhibits the highest hydrophilicity reflected by the lowest water contact angle (Supplementary Fig. 3). Notably, the weaker intensity of the P–O–H band in

MOPM-$Zr^{4+}$ accounts for its overshadowed hydrophilicity compared to MOPM-$Fe^{3+}$, which is in accordance with the decreased $HPO_4^-$ content from 19.6% to 12.3% caused by the more coordinated sites for $Zr^{4+}$ (Supplementary Fig. 4). The time-dependent dynamic water contact angle was recorded to further explore the hydrophilicity of the membrane surface (Fig. 1e). The water contact angle of PAN substrate decreases from instantaneous ~58.1° and stabilizes at ~28.4° after 540 s, showing typical behavior of the hydrophilic surface. Surprisingly, the instantaneous water contact angle is reduced to ~28.7° and the water drop could fully spread out to 0° in 29 s after loading MOPM-$Fe^{3+}$ on PAN substrate. This superhydrophilicity is highly advantageous for water transport across membrane.

Scanning electron microscope (SEM) images clearly show the divergent surface morphologies of the assembled MOPMs (Fig. 1f and Supplementary Fig. 5). The rougher surface of MOPM-$Fe^{3+}$ and MOPM-$Zr^{4+}$ embodies the drastic coordination reactions between PA and $Fe^{3+}/Zr^{4+}$, which is attributed to the phosphate group referring to the principle of hard base preferring hard acid (Supplementary Fig. 6). More importantly, the defects in MOPMs are gradually eliminated with the increased polarizing power of metal ions owing to the different coordination modes between metal ions and phosphate groups. The radiuses of metal ions are as follows: $Ag^+$ (0.126 nm), $Zn^{2+}$ (0.074 nm), $Ni^{2+}$ (0.069 nm), $Fe^{3+}$ (0.065 nm, high spin; 0.055 nm, low spin) and $Zr^{4+}$ (0.072 nm), respectively. Compared with other metal ions, the $Ag^+$ has the largest size and the lowest charge number, inducing polarization minimally and forming the longest and weakest metal-oxygen bond (1.895 Å), and thus bearing inferior coordination ability. Therefore, when acting as the central ion, the $Ag^+$ generates a looser structure with considerable defects than other metal ions. As the ionization potential of metal ions increases, the polarization intensifies the overlap of electron cloud and reduces the length of metal−oxygen bond from 1.879 Å to 1.575 Å (Supplementary Table 2). Also, owing to the enhanced charge-ability of metal ions, more deprotonated negatively-charged hydroxyl oxygen atoms in PA molecule are attracted and increase the coordinated sites for higher coordination degree, as verified by the decreased hydroxyl ratio in phosphate group from 24.6 to 12.3% (Supplementary Fig. 4). We also found that the PA/metal ratio decreases from 1:1.1 to 1:4.1 with the increased ionization potential (Supplementary Table 3), implying more central ions in MOPMs and leading to denser coordination network. As a result, when $Fe^{3+}$ or $Zr^{4+}$ with high ionization potentials serve as the central ions, the resultant MOPMs exhibits dense and defect-free structure. We chose MOPM-$Fe^{3+}$ for the subsequent investigation owing to its superhydrophilicity and defect-free structure favorable to water purification application.

**Structural optimization of ultrathin MOPMs.** One intriguing characteristic of MOPMs is their structural tunability depending on ligand concentration and metal ion content. As the PA concentration was reduced from 0.045 mg/mL to 0.030 mg/mL, the thickness of the resultant membrane decreased from 21.2 ± 0.2 nm to 13.2 ± 0.7 nm (Supplementary Figs. 7 and 8). By further decreasing the PA concentration to 0.015 mg/mL, a sub-10 nm membrane was achieved (Fig. 2a). The MOPM-$Fe^{3+}$ layer generated on the PAN substrate (Fig. 2b) is discerned from transmission electron microscopy (TEM) images and further determined as 8.3 ± 0.3 nm in thickness by atomic force microscope (AFM) height profile (Supplementary Fig. 7). After removing the PAN substrate by immersing the MOPM-$Fe^3$+/PAN membrane in N, N-dimethyl formamide (DMF), a transparent substrate-free MOPM-$Fe^{3+}$ membrane was obtained (inset in Fig. 2c and Supplementary Fig. 8) and could be

transferred onto silicon wafer. The coherent membrane structure keeps intact under microscale as demonstrated in Fig. 2c. In addition, the influence of $Fe^{3+}$ content on assembling MOPMs was monitored by the surface zeta potential measurements (Fig. 2d). An initial potential of −62.36 ± 0.41 mV for PAN substrate was detected, attributed to the partially-hydrolyzed nitrile groups into carboxyl groups. By absorbing PA molecules with abundant phosphate groups, a more negative surface potential was detected (−70.13 ± 0.44 mV), followed by an increase to −65.94 ± 0.51 mV after coordinated with positively-charged $Fe^{3+}$ ions. With the increase of $Fe^{3+}$ concentration, the electronegativity, as well as hydrophilicity of the membrane surface, decreases due to the more consumed hydroxyl groups by $Fe^{3+}$ (Supplementary Fig. 9). Interestingly, by using polyethylene glycol molecules with different $M_w$ as probe, we found that the effective mean pore size of MOPM-$Fe^{3+}$ decreases from 1.82 nm to 0.72 nm when the PA/Fe ratio is varied from 1:0.5 to 1:10, along with a narrowed pore size distribution (inset in Fig. 2d). The assembly behavior of $Fe^{3+}$-PA complex in solution was further explored to seek out the underlying principles governing the formation of the divergent porous structure of MOPMs depending on $Fe^{3+}$ content.

After mixing $Fe^{3+}$ and PA for 60 min, the electron transfer from donor (ligand) to acceptor (metal ion) induces the charge transfer transition and gives rise to an ultraviolet-range absorption at 275 nm, validating the formation of $Fe^{3+}$–PA coordination complexes (Fig. 2e), which is further confirmed by Raman spectra (Supplementary Fig. 10). The Tyndall effect indicates that the size of the complex is in the range of 1−100 nm (inset in Fig. 2e). Moreover, the PA/Fe ratio was found to substantially influence the aggregation behavior of complex. A higher PA/Fe ratio leads to a more turbid solution while a lower PA/Fe ratio leads to a more pellucid solution (inset in Fig. 2f) due to the corresponding larger $Fe^{3+}$–PA complex size up to 43.8 nm and smaller $Fe^{3+}$–PA complex down to 10.1 nm, respectively (Fig. 2f). The elucidation of the nucleation and structural evolution process of $Fe^{3+}$–PA complex at varied $Fe^{3+}$ contents as illustrated in Fig. 2g would provide a basis to better understand how the metal ion content influences membrane structure. During membrane formation, the PA molecules anchor on the PAN substrate, coordinate with $Fe^{3+}$ ions and generate $Fe^{3+}$–PA nuclei within 10 s, and the nuclei further grow up by coordinating with more ligands in solution and stacked into nanoporous layer (Supplementary Figs. 11 and 12). In the case of low $Fe^{3+}$ content (high PA/Fe ratio), each metal ion could bond with excessive PA molecules and the resultant unsaturated $Fe^{3+}$–PA complexes tend to aggregate into large $Fe^{3+}$–PA nuclei. While at high $Fe^{3+}$ content (low PA/Fe ratio), the ligand is in a relatively short supply and the PA molecule saturated by $Fe^{3+}$ would fail to bind each other, thus forming smaller $Fe^{3+}$–PA nuclei. During the thermal curing process in solution, the configuration change of PA molecules alleviates the steric constraint for $Fe^{3+}$ ion to generate highly coordinated and coherent membranes, and the stack-induced interstitial space would evolve into nanopores for separation via thermal shrinkage. Therefore, the nucleus size based on PA/Fe ratio determined the size of resultant nanopores in membrane and hence the structural compactness. At high PA/Fe ratio, the $Fe^{3+}$–PA nuclei tend to grow larger and stack together with larger interstitial space, thus resulting in larger nanopores in membrane with a loose structure, and vice versa. Based on this mechanism, we developed two kinds of sub-10 nm MOPM-$Fe^{3+}$ with typical loose and dense structures by facilely modulating PA/Fe ratio at 1:0.5 and 1:7, respectively (Fig. 2g). The high PA/Fe ratio (1:0.5) generates membrane with slightly higher thickness of ~9 nm (Supplementary Fig. 13) due to the larger $Fe^{3+}$–PA nuclei.

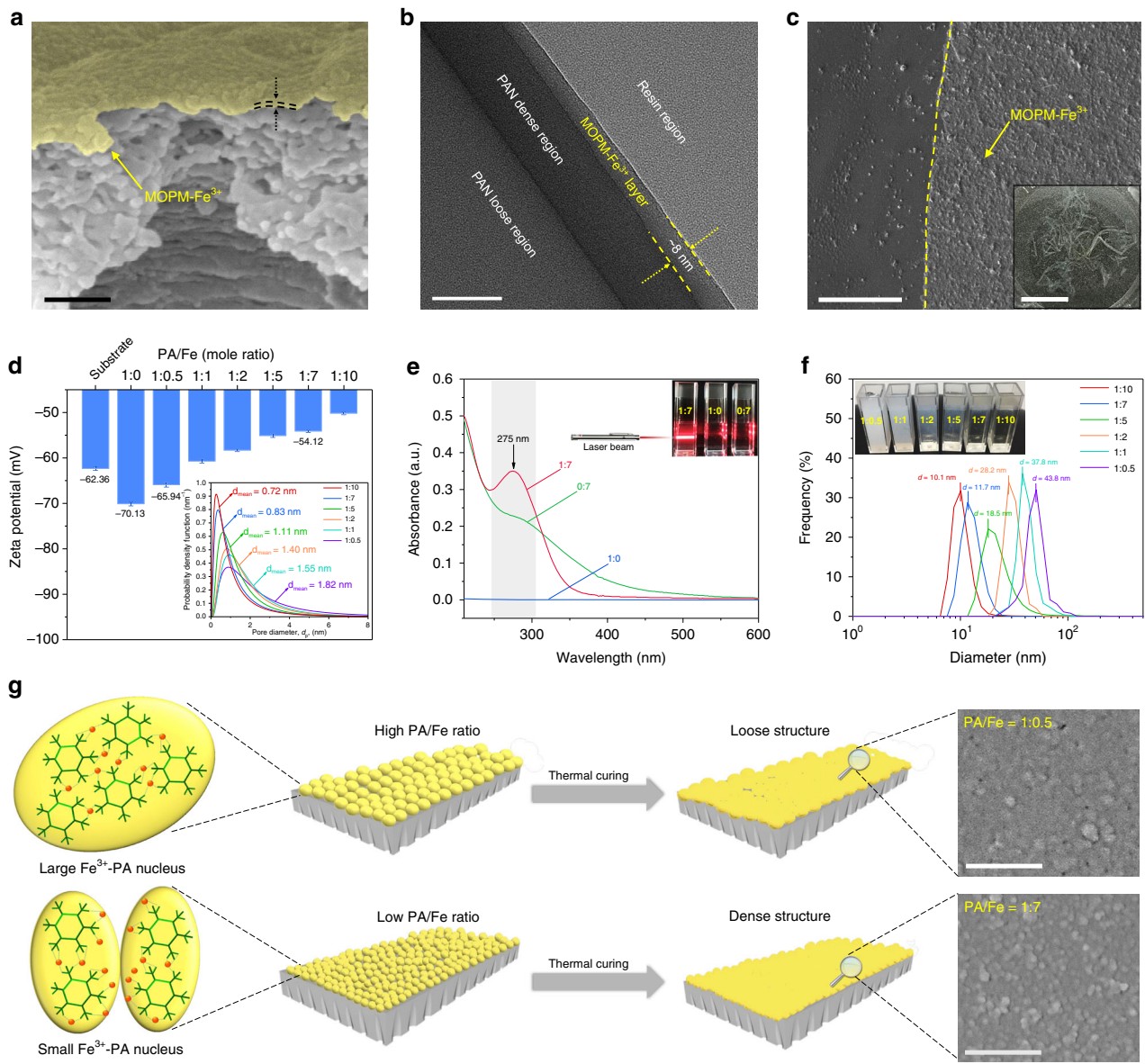

**Fig. 2** Optimizing assembly behavior for ultrathin MOPMs. **a** SEM images of MOPM-$Fe^{3+}$ on PAN substrate. False color of yellow was utilized to singularize the MOPM-$Fe^{3+}$. Scale bar: 100 nm. **b** TEM image of MOPM-$Fe^{3+}$ on PAN substrate. Scale bar: 20 nm. **c** Transferred MOPM-$Fe^{3+}$ on silicon wafer. Scale bar: 1 μm. Inset in **c**: Digital photograph of substrate-free MOPM-$Fe^{3+}$ in DMF. For **a–c**, the PA concentration was 0.015 mg/mL. **d** Surface zeta potential of membranes. Inset in **d**: Effective pore size distribution of MOPM-$Fe^{3+}$ with varied PA/Fe ratio using PEG as molecular probe. **e** Ultraviolet-visible spectra of aqueous solutions with different PA/Fe ratio. Inset in **e**: Photograph of aqueous solutions with different PA/Fe ratio under laser bean irradiation. **f** Size distribution of $Fe^{3+}$-PA complexes in assembly solution with varied PA/Fe ratio detected by dynamic light scattering. Inset in **f**: Digital photo images of MOPM-$Fe^{3+}$ complexes with varied PA/Fe ratio. **g** Schematic illustration of growth and formation for MOPM-$Fe^{3+}$ with high and low PA/Fe ratio. Inset in **g**: SEM images of MOPM-$Fe^{3+}$ with PA/Fe ratio of 1:0.5 (top) and 1:7 (bottom). Scale bar: 200 nm. Error bars represent standard deviations for 3 measurements

**Permselectivity and stability of MOPM/PAN membranes**. The permselective performance of resultant MOPM/PAN membranes was evaluated in a filtration apparatus (Supplementary Fig. 14) using five kinds of typical organic dyes with varied molecular size (0.74−2.20 nm) to simulate organic wastewater (Supplementary Fig. 15). The pristine PAN substrate and PA-treated PAN substrate showed low dye rejections (30−60%) owing to their large pores (Supplementary Fig. 16). As shown in Fig. 3a, the MOPM/ PAN membranes employing $Ag^+$, $Zn^{2+}$, and $Ni^{2+}$ as coordination metal ions still show inferior dye rejections similar to PAN

substrate due to the structural defects (Fig. 1e). Considerably higher dye rejections (88.5−100%) are achieved by using the metal ions ($Fe^{3+}$ and $Zr^{4+}$) with larger ionization potential which led to defect-free membrane structure. The permselective behavior of MOPM-$Fe^{3+}$/PAN membrane with potentially higher water permeance than the less hydrophilic MOPM-$Zr^{4+}$/PAN was further investigated.

The solute rejections of MOPM-$Fe^{3+}$/PAN are in an order of small-size dye (<1 nm, OG) < middle-size dye (~1 nm, RB)< large-size dye (> 2 nm, MB/CR/AB), indicating the size exclusion

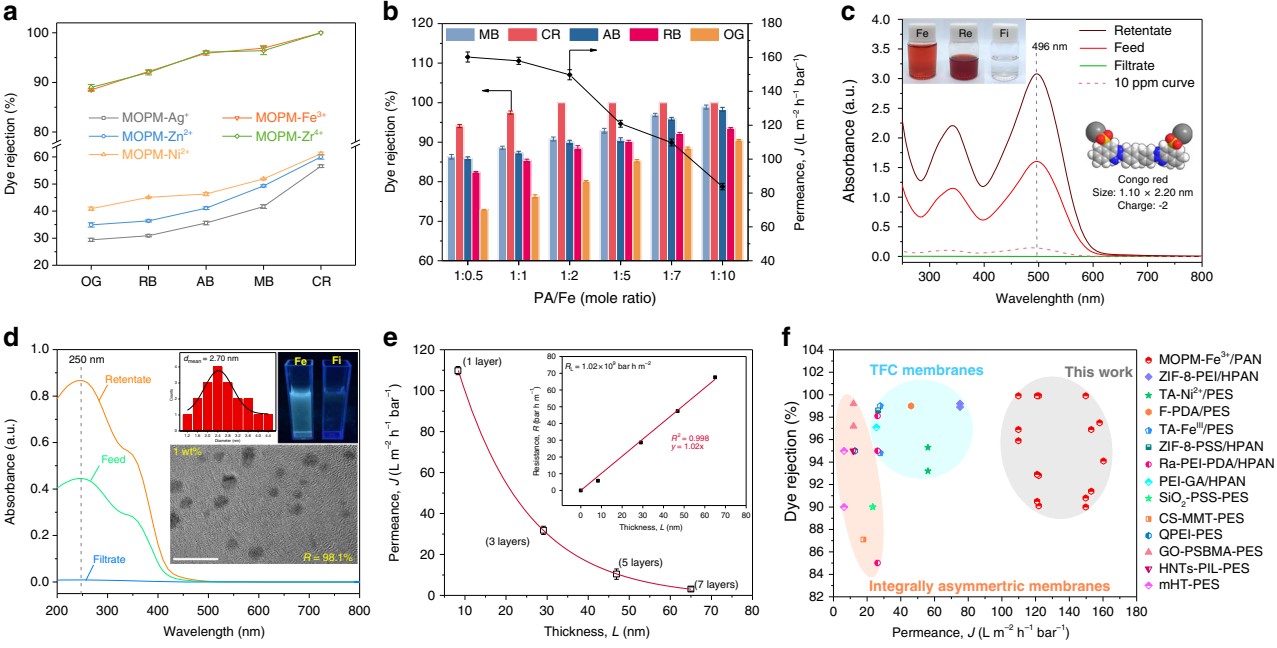

**Fig. 3** Permselectivity of MOPM/PAN membranes. **a**, **b** Filtration performance of MOPM/PAN membranes with different coordinated metal ions (**a**) and MOPM-$Fe^{3+}$/PAN membranes with varied PA/Fe ratio (**b**). 100 ppm of Methyl blue (MB, 1.62 × 2.03 nm), Congo red (CR, 1.10 × 2.20 nm), Alcian blue (AB, 1.42 × 2.20 nm), Rose Bengal (RB, 1.06 × 1.08 nm) and Orange GII (OG, 0.74 × 1.07 nm) solution as feed. **c** Ultraviolet-visible spectra of Congo red in feed, retentate, and filtrate. PA/Fe ratio = 1:7. Inset in **c**: Digital photo images of feed (Fe), retentate (Re) and filtrate (Fi) (top left) and molecular structure of Congo red (bottom right). **d** Ultraviolet-visible absorption spectra of graphene oxide quantum dots (GQDs) in feed, retentate, and filtrate. PA/Fe ratio = 1:0.5. Inset in **d**: Size distribution of GQDs (top left), digital photo images of feed (Fe) and filtrate (Fi) (top right) and TEM image of GQDs (bottom). Scale bar: 10 nm. **e** Water permeance of MOPM-$Fe^{3+}$/PAN membranes with different layer number of MOPM-$Fe^{3+}$. PA/Fe ratio = 1:7. Red line was the best exponential fit. Inset in **e**: Mass transport resistance as a function of thickness for MOPM-$Fe^{3+}$/PAN membranes with different skin layer thicknesses. Red line was the best linear fit. **f** Filtration performance of state-of-the-art polymeric membranes for water purification in literatures. Error bars represent standard deviations for 3 measurements

plays an important role during filtration. However, the influence of electrostatic interaction is a nonnegligible factor for the charged solute rejection of the negatively-charged MOPM-$Fe^{3+}$/PAN besides the size-based selectivity[31]. Specifically, even though CR, MB, and AB have similar size (1.10−2.20 nm), the rejections follow the order of CR > MB > AB due to their different charge property. The AB with positive charge prompts to be attracted by the negatively-charged MOPM-$Fe^{3+}$ skin layer and thus resulting in a higher solute permeace and hence lower rejection. As for the negatively-charged CR and MB, the membrane shows a higher rejection for CR with a more negative charge of −40.2 ± 0.7 mV than MB with a relatively less negative charge of −12.0 ± 0.6 mV. This indicates that the electrostatic repulsion contributes to high dye rejection. Moreover, the increased $Fe^{3+}$ content in assembly solution enhances the CR rejection of MOPM-$Fe^{3+}$/PAN gradually from 94.1% to 100% (Fig. 3b) along with the decreased permeace due to the denser membrane structure (inset in Fig. 2d). Figure 3c shows that the ultrathin and dense MOPM-$Fe^{3+}$/PAN (PA/Fe ratio = 1:7) with high water permeace of ~109.8 L m$^{-2}$ h$^{-1}$ bar$^{-1}$ could completely reject CR in feed (100 ppm) and yield purified water (~0 ppm) and concentrated retentate. As shown in Supplementary Figs. 17 and 18, the membrane also exhibits high rejections for AB (95.9%) and MB (96.9%) as well as moderate rejection for RB (92.5%) and OG (88.5%). Besides dye solution, we also explored the separation potential of the loose MOPM-$Fe^{3+}$/PAN (PA/Fe ratio = 1:0.5) in nanoparticle removal and found that it showed a fairish rejection of 98.1% for graphene oxide quantum dots ($d_{mean}$ = 2.70 nm) in aqueous solution (Fig. 3d) resulting

from its suitable nanopore size ($d_{mean}$ = 1.82 nm). Under this premise, its superior permeace of ~160.3 L m$^{-2}$ h$^{-1}$ bar$^{-1}$ would make it available to remove nanoadsorbents from wastewater.

Membrane thickness is generally related to the length of mass transport pathway across membrane and positively proportional to mass transport resistance (Supplementary Table 4). To clarify the structure-performance relation between membrane thickness and mass transport resistance, the MOPM-$Fe^{3+}$/PAN membranes with 1−7 skin layers were prepared through repeated layer-by-layer (LBL) assembly (Supplementary Fig. 19). The membrane permeace sharply decreases from ~109.8 L m$^{-2}$ h$^{-1}$ bar$^{-1}$ to ~14.1 L m$^{-2}$ h$^{-1}$ bar$^{-1}$ with the skin layer thickness ($L$) increasing approximately from ~8 nm (1 layer) to ~65 nm (7 layers) and corresponding resistance ($R$) from 5.78 bar h m$^{-1}$ to 67.62 bar h m$^{-1}$ (Fig. 3e). This can be explained by solution-diffusion mechanism for water transport across dense membrane[31]. Since the LBL assembly has little influence on the hydrophilicity of membrane (inset in Supplementary Fig. 19), the change in water solution coefficient is negligible. Nonetheless, the diffusion coefficient decreases owing to the prolonged pathway for dissolved water molecules to diffuse across. By linearly fitting the resistance with membrane thickness, the thickness-independent resistance ($R_L$) of MOPM-$Fe^{3+}$ was determined to be about 1.02 × 10$^9$ bar h m$^{-2}$ ($R^2$ = 0.998), suggesting the homogeneous structure of these LBL-assembled MOPM-$Fe^{3+}$ membranes.

Insights into water transport across membrane can be obtained in terms of Arrhenius activation energy ($E_a$)[32], which reflects the

synergetic resistance originated from physical structure and chemical affinity of membranes. An $E_a$ value of 9.56 kJ mol$^{-1}$ for MOPM-Fe$^{3+}$/PAN (~8 nm) was obtained by fitting the permeation rate and operating temperature, whereas this index increased to 13.39 kJ mol$^{-1}$ when the skin layer thickness increased to ~65 nm (Supplementary Fig. 20). This indicates that the water diffusion across thicker skin layer becomes the rate-limiting step and can be activated by elevating temperature while the water diffusion across thinner skin layer is less sensitive to temperature. Furthermore, it is interesting to find that the $E_a$ value (9.56 kJ mol$^{-1}$) of MOPM-Fe$^{3+}$/PAN is about 60−70% lower than that of traditional polymeric membranes such as alginate polymer (31.29 kJ mol$^{-1}$)[33] and polyamide (23.80 kJ mol$^{-1}$)[34], and even lower than that of graphene oxide membranes (11.12 kJ mol$^{-1}$)[19] and aquaporin-incorporated polymeric membranes (AqpZ, 14.23 kJ mol$^{-1}$)[4] which bear ultrafast water channels. The efficient water transport through the metal-organophosphate membrane is largely attributed to the sub-10 nm thickness and superhydrophilic nature of the MOPM-Fe$^{3+}$ layer. This backs up the conceptualized design of biomimetic water channel composed of hydrophilic entrance rims and low-resistance physical architecture[4]. As a result, the MOPM-Fe$^{3+}$/PAN membranes exhibit unprecedented permeance in the range of 109.8−160.3 L m$^{-2}$ h$^{-1}$ bar$^{-1}$, remarkably higher than that of previously reported polymeric TFC and integrally asymmetric membranes with comparable dye rejections (Fig. 3f and Supplementary Table 5).

Beyond water flux, membrane fouling is another notorious bottleneck for water-purification membranes by causing decreased flux, frequent cleaning and shortened lifetime. Superhydrophilic surface like betaine-phosphate-enriched cell membrane could alleviate fouling by forming hydration layer to resist foulant adhesion[35]. Similarly, the distinctive superhydrophilicity of MOPM-Fe$^{3+}$ brings about outstanding fouling resistance to the three major categories of typical organic foulants including sodium alginate (SA), humid acid (HA) and bovine serum albumin (BSA), representing polysaccharide, natural organic matter and protein, respectively (Fig. 4a). The permeation recovery ratios of MOPM-Fe$^{3+}$/PAN membrane for three kinds of foulants are all over 90% in the first cycle and reach nearly 100% in the following 4 cycles (Supplementary Fig. 21). The BSA absorption measurement verifies the lower amount of BSA adhered to MOPM-Fe$^{3+}$/PAN surface under both dynamic and static condition compared with PAN substrate (Supplementary Fig. 22), accounting for its superior antifouling performance during filtration. Besides, the electrostatic repulsion between both negatively-charged foulants (SA, BSA, and HA) and MOPM-Fe$^{3+}$/PAN membrane surface also contribute to the higher antifouling performance as compared to the positively-charged foulants (Supplementary Fig. 23).

For practical membrane-based filtration, the existence of acid or base components in wastewater and the use of acid or base reagents for membrane cleaning demand high stability of membranes against varied pH conditions. The binding intensity among membrane-forming components determines the overall durability of membrane and the coordination between the strongly electron-donating organophosphate ligand and

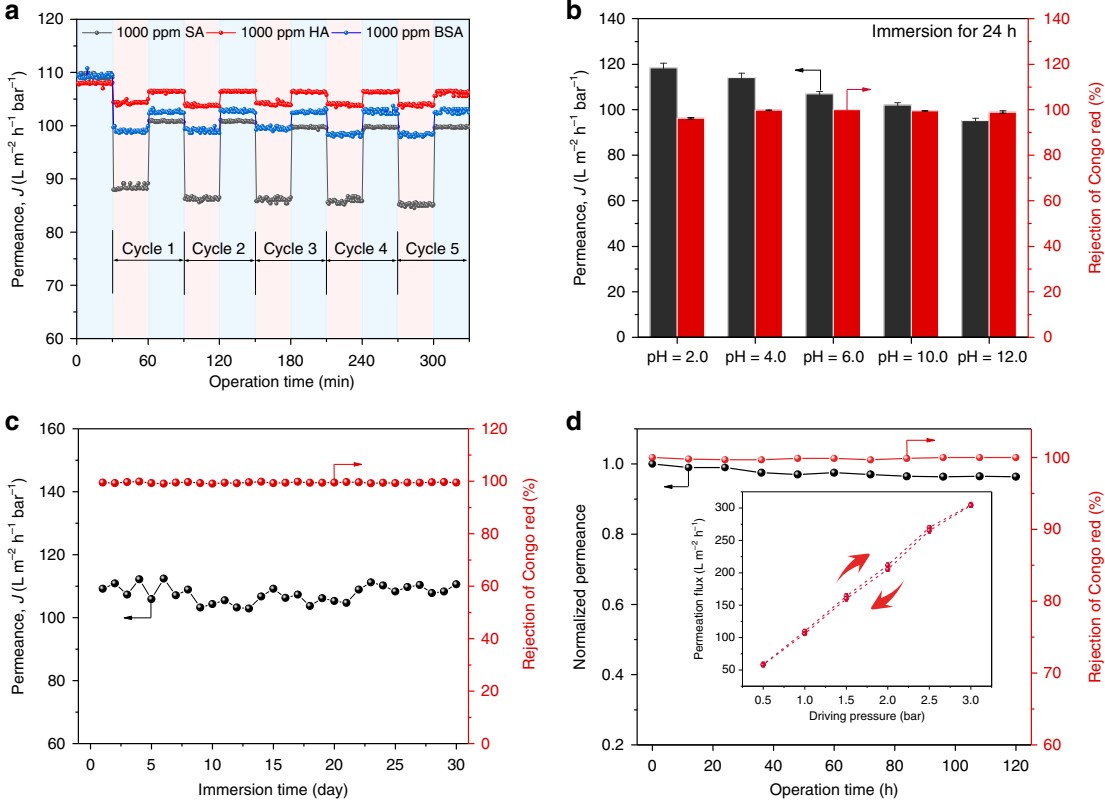

**Fig. 4** Stability of MOPM-Fe$^{3+}$/PAN membranes. **a** Five-stage antifouling measurement of MOPM-Fe$^{3+}$/PAN membrane with 1000 ppm of sodium alginate (SA), humid acid (HA) and bovine serum albumin (BSA) solution as feed at 1.0 bar. **b** Stability performance of MOPM-Fe$^{3+}$/PAN membrane under varied pH conditions by immersing membrane in HCl and NaOH solution for 24 h. T = 18 °C. **c** Long-term water stability of MOPM-Fe$^{3+}$/PAN membrane (pH = 4.0). **d** Long-term filtration performance of MOPM-Fe$^{3+}$/PAN membrane at 0.5 bar. Inset in **d**: Permeation flux of MOPM-Fe$^{3+}$/PAN membrane in pressure cycling experiment. Error bars represent standard deviations for 3 measurements

multivalent metal ions has been proved to be stable in acid solution[24]. We have demonstrated the strong Fe–O–P binding with a binding energy of $-315.1$ kJ mol$^{-1}$, in accordance with its high isotropic hyperfine-coupling constant comparable to covalent bond[36]. This strong-coordination bond makes MOPM-Fe$^{3+}$ highly stable. After immersion in solution with pH ranging from 2.0 to 12.0 for 24 h, no defects are observed on membrane surface (Supplementary Fig. 24) and the CR rejection remains as high as 96.2% up to 100% (Fig. 4b). The alkali-facilitated deprotonation of phosphate enhances the Fe$^{3+}$–PA coordination and increases the compactness of metal-organic networks[23], thus slightly decreasing water permeance, and vice versa. Especially, as shown in Fig. 4c, the MOPM-Fe$^{3+}$ displays superior stability during one-month water immersion (pH = 4.0). This demonstrates the distinctive acid stability of MOPM-Fe$^{3+}$ compared to conventional metal-polyphenolic networks. From the perspective of analytical chemistry, the stability constant of Fe$^{3+}$-PA complex ($K = 1.6 \times 10^{18}$)[37] is nearly nine orders of magnitude higher than that of Fe$^{3+}$–TA complex ($K = 3.4 \times 10^{9}$)[38] under acidic condition (pH = 5.0). This is because the phenolic hydroxyl bearing p–π conjunction of electrons with benzene ring served as weak electron donor for metal ion, whereas the low-electronegativity phosphorus and feedback electrons from double-bonded oxygen synergistically enhanced the electron-donating ability of hydroxyl oxygen in phosphate group. Furthermore, the 5-day filtration test for MOPM-Fe$^{3+}$/PAN membrane confirms its long-term operation stability, and the near-linear relation between permeation flux and driving pressure (0.5−3.0 bar) during pressure cycling experiment reflects the satisfactory compaction resistance of membrane (Fig. 4c). Besides, the strong Fe–O–P binding also brings about structural stability under saline conditions (Supplementary Fig. 25), low metal leakage during filtration (Supplementary Table 6) and sufficient mechanical strength (Supplementary Fig. 26). The comprehensive stability in various harsh conditions makes MOPM-Fe$^{3+}$/PAN membrane a promising candidate for practical water purification.

## Discussion

In summary, we proposed a strong-coordination-based assembly strategy for fabrication of sub-10 nm metal-organophosphate membranes (MOPMs). The metal-organophosphate coordination with moderately strong intensity between covalent and non-covalent interactions enabled the controllable formation of robust membrane structure under mild conditions in aqueous solution. By altering metal ion types (Ag$^+$, Zn$^{2+}$, Ni$^{2+}$, Fe$^{3+}$, and Zr$^{4+}$), we found that the Fe$^{3+}$ and Zr$^{4+}$ ions with stronger electron acceptability and thus higher binding energy or ionization potential are conducive to fabricate defect-free membranes while MOPM-Fe$^{3+}$ bearing superhydrophilicity is more preferred choice. Impressively, we tuned the MOPM-Fe$^{3+}$ membrane in thickness to 8-nm through reducing the ligand concentration to 0.015 mg/mL. By further manipulating the nucleation and growth of PA–Fe$^{3+}$ complexes, denser porous structure could be achieved at low PA/Fe ratio (1:7). The ultrathin thickness and ultrahigh water affinity of MOPM-Fe$^{3+}$ afforded much lower energy barrier for water permeation. The membrane comprising the 8-nm, superhydrophilic and dense MOPM-Fe$^{3+}$ skin layer on PAN substrate exhibits a water permeance of 109.8 L m$^{-2}$ h$^{-1}$ bar$^{-1}$ with rejections of >95% to various dyes (molecular size >2 nm), yielding an overall water purification efficiency beyond the state-of-the-art polymeric membranes. Especially, the MOPM-Fe$^{3+}$ demonstrates one-month stability in acid aqueous condition (pH = 4.0) owing to the strong Fe–O–P bonds. This facile and controllable assembly strategy based on moderately strong molecular interaction could enable scalable, green and low-cost ultrathin membrane manufacturing and realize energy-saving water resource reclamation.

## Methods

**Materials.** Materials and reagents used in this work are given in the Supplementary Methods.

**Assembly of MOPMs.** Firstly, the PAN substrate was immersed in ethanol followed by air drying to remove contaminant. Secondly, the PAN substrate was facedown submerged in PA solution (25.5 mL) for 5 min. Sequently, the solution with transition metal ion (M$^{n+}$, 4.5 mL) was poured into PA solution to trigger metal-organic assembly for a schedule time. Afterward, the solution and PAN substrate coated by MOPMs was transferred into oven for thermal curing at 60 °C for 10 min. Finally, the MOPM/PAN composite membrane was rinsed with DI water (pH = 6.0 ± 0.2) for 10 min to remove most residual weakly bound PA molecules and metal ions and stored in DI water before use. The as-prepared membrane was denoted as MOPM-M$^{n+}$, where the M$^{n+}$ represented Ag$^+$, Zn$^{2+}$, Ni$^{2+}$, Fe$^{3+}$, and Zr$^{4+}$. The details are described in Supplementary Methods (Supplementary Table 7).

**Simulation.** The coordination between transition metal ions (Ag$^+$, Zn$^{2+}$, Ni$^{2+}$, Fe$^{3+}$, and Zr$^{4+}$) and organophosphate (methyl phosphate) was carried out with Material Studio software through the first-principles density functional theory (DFT) approach. The phosphate group was set to be completely deprotonated for coordination. All major calculations were carried out using DMol3 module with the basis set of dynamic nuclear polarization (DNP) and the function of generalized gradient approximation (GGA) and Perdew-Wang 91 (PW91)[39]. The binding energy ($\Delta E_{binding}$) between methyl phosphate and metal ion was calculated as follows:

$$\Delta E_{binding} = E_{(phosphate/metal\,ion)} - E_{(methyl\,phosphate)} - E_{(metal\,ion)} \quad (1)$$

**Characterization.** SEM images were obtained from a Nanosem 430 field-emission scanning electron microscopy. TEM was utilized to capture cross-section image of MOPM-Fe$^{3+}$/PAN composite membrane. AFM images were taken from a Bruker Dimension Icon atomic force microscopy. FTIR spectra were taken from a Nicolet 560 Fourier transform infrared spectroscopy. Water contact angles were measured on a Data-Physics OCA 15EC with 4 μL water droplet. EDX mapping images were taken from a Genesis XM2 APEX 60SEM energy dispersive X-Ray spectroscopy. The size of Fe$^{3+}$-PA complex was measured by a Zetasizer Nano ZS90 dynamic light scattering. Surface zeta potential was obtained from a SurPASS Electrokinetic Analyzer. XPS was taken from a ESCALAB 250Xi X-Ray photoelectron spectroscopy. UV-vis spectra were taken from a Hitach UV-3010 ultraviolet-visible spectrophotometer. Chemical oxygen demand (COD) was detected by Leichi COD-571. The pore size distribution of membranes was determined by rejection experiments using PEG with different molecular weight ($M_w = 200, 400, 600, 1000, 2000, 4000,$ and 6000 Da) at concentration of 50 ppm as feed solution. The specification is illustrated in the Supplementary Methods.

**Filtration performance measurements.** Membrane performance was evaluated using a dead-end stirred cell filtration apparatus (Amicon 8010) with an effective area of 4.1 cm$^2$. Rejection experiments were conducted with model solution (Methyl blue, Congo red, Alcian blue, Rose Bengal, Orange GII and GQDs). The permeance ($J$, L m$^{-2}$ h$^{-1}$ bar$^{-1}$) and solute rejection ratio ($R$, %) were calculated by following equations (Eq. 2 and Eq. 3):

$$J = \frac{V}{A\Delta t \Delta p} \quad (2)$$

$$R = \frac{C_f - C_p}{C_f} \times 100\% \quad (3)$$

where $V$ (L) was the volume of the permeate, $A$ (cm$^2$) was the effective membrane area, $\Delta t$ (h) was the permeating time and $\Delta p$ is the driving pressure. $C_p$ (ppm) and $C_f$ (ppm) were the solute concentration in permeate and feed solutions, respectively. Solute concentrations were determined with a UV-vis spectrophotometer and the 10 ppm dye curves were measured as references for accurate rejection value for dyes. The concentrated CR retentate was diluted ten times to accurately measure concentration. The specific description is shown in the Supplementary Methods. The mass transfer resistance ($R$, bar h m$^{-1}$) of membrane was calculated by following equations (Eq. 4, Eq. 5 and Eq. 6):

$$J_{TFC} = \frac{1}{R_{substrate} + R_{MOPM}} \quad (4)$$

$$J_{substrate} = \frac{1}{R_{substrate}} \quad (5)$$

$$R_{\text{MOPM}} = \frac{1}{J_{\text{TFC}}} - \frac{1}{J_{\text{substrate}}} \tag{6}$$

where $J$ (L m$^{-2}$ h$^{-1}$ bar$^{-1}$) was the permeance of membrane and $J_{\text{substrate}}$ was measured to be ~300 L m$^{-2}$ h$^{-1}$ bar$^{-1}$.

**Stability performance measurements.** Antifouling performance of membrane was measured by filtration experiment using sodium alginate (SA), humid acid (HA) and bull serum albumin (BSA) as model organic foulants. The pH stability of membrane was conducted by immersing membrane in HCl or NaOH solution (pH = 2.0–12.0) for 24 h followed by performance measurement. The compaction resistance of membrane was evaluated by measuring water permeation flux (L m$^{-2}$ h$^{-1}$) at different driving pressure (0.5–3.0 bar). The long-term operation stability of membrane was tested by filtrating Congo red solution (100 ppm) for 5 days. The long-term water stability of membrane was measured by immersing membrane in water (pH = 4.0) followed by filtration test. The specific description is shown in Supplementary Methods.

## Data availability

The source data underlying Figs. 1b–d, 2d–f, 3a–f and 4a–d and Supplementary Figs. 1a, b, 3, 4b–h, 7a, c, e, 9a, b, 10, 16a–c, 17a, b, 18a–f, 20a–d, 21a, b, 22, 23, 25e–h are provided as a Source Data file. The data that support the findings of this study are available from the corresponding author upon reasonable request.

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

## Acknowledgements

The authors gratefully acknowledge financial support from National Natural Science Foundation of China (21878215, 21621004, 21576189, 21490583), National Science Fund for Distinguished Young Scholars (21125627), Natural Science Foundation of Tianjin City (18JCZDJC36900) and National Key Research and Development Plan (2017YFB0603400).

## Author contributions

H.W., Z.J., and X.Y. conceived the idea and designed the research. X.Y. and K.X. carried out the experiment. Q.Y. and J.Y. performed water contact angle measurement. X.Y. and M. H. carried out the simulation. L.C. provided constructive suggestions for results and discussion. Y.S. and R.Z. helped to revise the paper. All authors participated in discussion. H.W., Z.J., X.Y., L.C., R. Z., and Y.S. co-wrote the paper.

## Additional information

**Competing interests:** The authors declare no competing interests.

