## [Peer Review File · Nature Communications]

Reviewers' comments:

Reviewer #1 (Remarks to the Author):

This manuscript reports a new series of ultrathin film composite membranes formed by self-assembly of phytic acid (electron donor) and metal ions (electron acceptor). The selective layer can be less than 10 nm without defects for water purification. The effect of the acid strength of the metal ions (Lewis acid) on the desalination performance is investigated. These materials are new to the membrane field and should be interesting for the broad community of surface science, membrane science, and water purification. My specific comments are shown below.

1. I suppose that the selective layer is similar to the cation exchange membranes, due to the fixed anions and mobile protons. Would this be a concern if the dyes are positively charged because they can be easily adsorbed on the membrane surface?
2. The charged membranes may have high hydrophilicity. But they may not have good antifouling properties, especially if the feed water contains foulants with countercharges. The authors should make this clear to the readers.
3. Dye solutions may contain a high content of salts such as Na_2SO_4 and NaCl . While the Fe ions have strong binding energy with the phytic acids, the Fe ions can still be substituted by Na^+ or others, leading to loosed structure. How would this affect the long term stability of the membranes?
4. What would happen to the membranes if the wastewater has high PH values? For example, if NaOH is presented, would it neutralize the phytic acid and change the nanostructure of the membranes?

Reviewer #2 (Remarks to the Author):

This manuscript reports metal organophosphate membranes (MOPMs) fabricated by mixing natural phytic acid (PA) with heavy metal ions (including Ag^+ , Zn^{2+} , Ni^{2+} , Fe^{3+} , and Zr^{4+}) for nanofiltration. Five MOPM-metal ion membranes were prepared on PAN substrate, it seems that only MOPM- Fe^{3+} membrane could have a thickness down to sub-10 nanometers, and high water permeance up to $109.8 \text{ L m}^{-2} \text{ h}^{-1} \text{ bar}^{-1}$, high rejection (>95 %) of dyes with molecular sizes > 2 nm. The authors also claim that MOPM- Fe^{3+} membranes have very good stability and regenerability because of the stronger metal-organic coordination between metal ions and organophosphate ligands. Therefore, I believe that this manuscript would be potentially accepted in Nature Communications after some revisions.

1. The first major comment is the "sub-10 nm" in the manuscript title, which is kind of confusing. It's difficult to tell whether the "sub-10 nm" refers to the pore size or the thickness of the membrane without any context.
2. The second major comment is about the membrane fabrication. The authors claim that the self-assembled molecular layer is formed by H-bond interaction between the carboxyl groups on hydrolyzed PAN and the phosphate groups on PA. Does the pH value of the PA solution effect on the membrane quality and thickness. As we know, if the solution pH value is higher than the isoelectric points of the PAN and PA, both the carboxyl groups and phosphate groups are negatively charged, and there should be electrostatic repulsion between them, which is adverse to the formation of PA layer on the substrate. In addition, the metal ions were added subsequently into the PA solution, it is possible that the metal ions would coordinate with carboxyl groups on PAN firstly, serving as cross-linkers, and then coordinate with phosphate groups via the unsaturated metal sites.
3. The third major comment is about the fabrication process of using heavy metal ions to joint

PAs, which was carried out at 60 °C. Is the temperature essential for the final quality of the MOPM membranes? It seems that 60 °C is only suitable for fabricating high-quality the MOPM-Fe³⁺/Zr⁴⁺ membranes as shown in Figure 1f. Therefore, more details should be given in the supporting why the authors cannot get defect-free membranes based on other three metal ions under this condition.

4. The fourth major comment is about the dyes used for rejection testing and the mechanism for the high dye rejection. The selected dyes for rejection testing are methyl blue (1.6×2.0 nm), congo red (1.1×2.2 nm), and alcian blue (1.4×2.2 nm). They have very close molecular sizes, but the MOPM-Fe³⁺ membrane showed the highest rejection for congo red as shown in Figure 4. Why the authors chose these three dyes? How about the membrane performance for filtration much smaller organic molecules?

As shown in Figure S 14. There are sulfonate groups on methyl blue and congo red, and positive charges on alcian blue. As a result, the negatively charged MOPM-Fe³⁺/PAN membranes may reject the negatively charged methyl blue and congo red molecules by electrostatic repulsion, while adsorb positively charged alcian blue molecules through electrostatic attraction onto the negatively membrane surface or into the pores of the membranes, showing the lowest rejection of alcian blue. Therefore, a size sieving effect is not suitable to explain the rejection mechanism of the MOPM membranes. Some experimental evidence should be provided to confirm the adsorption and repulsion effects can be ignored if the authors think the size sieving effect is the main reason for dye rejection.

4. Even though the filtration performance of MOPM-Fe³⁺/PAN membranes is remarkable, the performance of pristine PAN and PA treated PAN should be included as comparative experiments.

5. For practical membrane-based filtration, the high water permeance, long-term stability and anti-fouling ability really matter. Meanwhile, the membrane mechanical strength should be considered.

6. Fig S2 in the supporting information cannot be seen clearly. High quality image should be provided.

7. Since the authors used the MOPMs fabricated based on heavy metal ions for water treatment, it would be better to provide some data of the amount of heavy metal ions in the permeate water.

A detailed response to the reviewers' comments

Reviewer #1 (Remarks to the Author):

This manuscript reports a new series of ultrathin film composite membranes formed by self-assembly of phytic acid (electron donor) and metal ions (electron acceptor). The selective layer can be less than 10 nm without defects for water purification. The effect of the acid strength of the metal ions (Lewis acid) on the desalination performance is investigated. These materials are new to the membrane field and should be interesting for the broad community of surface science, membrane science, and water purification. My specific comments are shown below.

Reply:

Thank the reviewer for the highly positive comments.

1. I suppose that the selective layer is similar to the cation exchange membranes, due to the fixed anions and mobile protons. Would this be a concern if the dyes are positively charged because they can be easily adsorbed on the membrane surface?

Reply:

Thank the reviewer for the valuable comment. Indeed, most of membranes and organic dyes are charged in aqueous solution (*J. Mater. Chem. A*, 2018, 6: 13331-13339; *J. Mater. Chem. A*, 2018, 6: 13191-13202). Both negatively-charged and positively-charged membranes will tend to adsorb dyes with countercharges. Besides, other weak interactions such as H-bond and Van der Waals force between dye molecules and membrane surface may also trigger the adsorption. Therefore, adsorption is a common phenomenon when filtrating dye wastewater and should be carefully considered. In this work, we investigated the adsorption of five kinds of dyes including the **negatively-charged** Methyl Blue (1.62×2.03 nm), Congo Red (1.10×2.20 nm), Rose Bengal (1.06×1.08 nm), Orange GII (0.74×1.07 nm) and the **positively-charged** Alcian Blue (1.42×2.20 nm) on the MOPM-Fe³⁺/PAN membrane. The structural information and charge property of the dyes are shown in **Figure R1** and the adsorption amounts are shown in **Figure R2**.

Figure R1. Structural information of organic dyes used in this work.

Figure R2. The dye adsorption on MOPM-Fe³⁺/PAN membrane.

The adsorption amounts of Methyl blue, Congo red, Alcian blue, Rose Bengal and Orange GII are 16.5±1.0 µg/cm², 16.0±0.8 µg/cm², 17.2±1.0 µg/cm², 16.3±1.0 µg/cm² and 12.9±0.6 µg/cm², respectively. Among the Methyl blue, Alcian blue and Congo red with similar size, the positively-charged Alcian blue shows higher adsorption than Methyl blue and Congo red, which accounts for its lower rejection. Fortunately, the Alcian blue features large molecular size (1.42×2.20 nm). As a result, the membrane still exhibits high rejection for the Alcian blue (95.9%). In general, the pore size of membrane should be appropriately manipulated to alleviate the influence of the electrostatic attraction between the organic dyes and membrane surface for sufficient dye rejections.

The adsorption data of various dyes on membrane has been added in the **Supplementary Material** as follows.

The dye adsorption was measured by fixing membrane into filtration cell (effective area=1.77 cm²) with 10 mL of feed solution (100 ppm dye). After 3-day adsorption, the dye concentration of feed was determined by ultraviolet-visible spectrophotometer and the amount of adsorbed dyes (M_{dye} ,

$\mu\text{g}/\text{cm}^2$) on the membrane was calculated by following equation (Eq. S6)

$$M_{\text{dye}} = \frac{V_S}{A} \times (C_0 - C_3) \quad (\text{S6})$$

where the C_0 , C_3 ($\mu\text{g}/\text{mL}$) were the dye concentration at the beginning and after 3 days, respectively, V_S (mL) was the volume of solution, the A (cm^2) was the effective membrane area. The reported data were the mean values of triplicate samples for each membrane.

Supplementary Figure 18. Ultraviolet-visible absorption spectra of (a) Methyl blue (MB), (b) Alcian blue (AB), (c) Rose Bengal (RB), (d) Orange GII (OG) and (e) GQDs in feed and filtrate of MOPM-Fe³⁺/PAN membrane. Inset: Digital photo images of feed (Fe) and filtrate (Fi) (top left) and molecular structure of MB and AB. The filtrate of cycle 1 (5 mL feed) was used as the feed of cycle 2. (f) The dye adsorption of MOPM-Fe³⁺/PAN membrane. The PA concentration, PA/Fe ratio and assembly time were fixed at 0.015 mg/mL, 1:7 and 60 min, respectively.

2. The charged membranes may have high hydrophilicity. But they may not have good antifouling properties, especially if the feed water contains foulants with countercharges. The authors should make this clear to the readers.

Reply:

Thank the reviewer for the valuable comments. The common organic foulants in wastewater include polysaccharides, natural organic matters and proteins, represented by sodium alginate, humic acid and bull serum albumin, respectively. They are all negatively charged in water with near neutral pH values and usually used as model foulants to evaluate the antifouling property of membranes (*Nanoscale*, 2017, 9: 7508-7518; *Ind. Eng. Chem. Res.*, 2018, 57: 4430-4441; *J. Membr. Sci.*, 2017, 540: 454-463). The negatively-charged membranes often exhibit good resistance to the above foulants due to electrostatic repulsion. To address the reviewer's concern, we tested the antifouling performance of MOPM-Fe³⁺/PAN membrane towards the positively-charged lysozyme as the model foulant (*J. Membr. Sci.*, 2018, 564: 788-799) as shown in **Figure R3**. The membrane exhibited deteriorated permeance for the lysozyme solution due to the electrostatic attraction between positively-charged foulant and negatively-charged membrane surface.

Figure R3. Five-stage antifouling measurement of MOPM-Fe³⁺/PAN membrane with 1000 ppm of lysozyme as feed at 1.0 bar.

The relevant information has been added in the revised **manuscript** and **Supplementary Material**

as follows.

Beyond water flux, membrane fouling is another well-recognized bottleneck for water-purification membranes by causing decreased flux, frequent cleaning and shortened lifetime. Superhydrophilic surface like betaine-phosphate-enriched cell membrane could alleviate fouling by forming hydration layer to resist foulant adhesion³⁵. Similarly, the distinctive superhydrophilicity of MOPM-Fe³⁺ affords outstanding fouling resistance to the three major categories of typical organic foulants including sodium alginate (SA), humic acid (HA) and bovine serum albumin (BSA), representing polysaccharide, natural organic matter and protein, respectively (Fig. 4a). The permeation recovery ratios of MOPM-Fe³⁺/PAN membrane for three kinds of foulants are all over 90% in the first cycle and reach nearly 100% in the following 4 cycles (Supplementary Figure 21). The BSA adsorption measurement verifies the lower amount of BSA adhered to MOPM-Fe³⁺/PAN surface under both dynamic and static condition compared with PAN substrate (Supplementary Figure 22), accounting for its superior antifouling performance during filtration. Besides, the electrostatic repulsion between the both the negatively-charged foulants (SA, BSA and HA) and the MOPM-Fe³⁺/PAN membrane surface also contribute to the higher antifouling performance as compared to the positively-charged foulants (Supplementary Figure 23).

Supplementary Figure 23. Five-stage antifouling measurement of MOPM-Fe³⁺/PAN membrane with 1000 ppm of lysozyme as feed at 1.0 bar. The PA concentration, PA/Fe ratio and assembly time were 0.015 mg/mL, 1:7 and 60 min, respectively.

3. Dye solutions may contain a high content of salts such as Na_2SO_4 and NaCl . While the Fe ions have strong binding energy with the phytic acids, the Fe ions can still be substituted by Na^+ or others, leading to loosed structure. How would this affect the long term stability of the membranes?

Reply:

Just as the reviewer pointed out, dye solutions may contain a high content of salts such as Na_2SO_4 and NaCl . To investigate the stability of the membrane in salt-containing dye solutions, we have performed some molecular dynamic simulation and experiments. Firstly, we evaluated the binding intensity between phosphate group and $\text{Na}^+/\text{Fe}^{3+}$ ions *via* molecular dynamic simulation. As shown in **Figure R4**, the binding energy between Fe^{3+} and methyl phosphate (-315.1 kJ/mol) is nearly 6 times higher than that between Na^+ and methyl phosphate (-45.8 kJ/mol). This is because the Na^+ ion interacts with the deprotonated phosphate group by electrostatic interaction while the Fe^{3+} ion interacts with the deprotonated phosphate group by strong coordination bond.

Figure R4. The simulated binding mode between metal ion ($\text{Na}^+/\text{Fe}^{3+}$) and methyl phosphate.

Then, we evaluated the long-term stability of MOPM- Fe^{3+} /PAN membrane under saline condition with Na_2SO_4 and NaCl concentration of 1000 ppm which is a commonly used concentration in literatures (*J. Membr. Sci.*, 2017, 540:391-400; *J. Membr. Sci.*, 2017, 539:52-64). As shown in **Figure R5**, after immersion for 2 weeks, the structure of MOPM- Fe^{3+} /PAN membrane remains intact. And the filtration performance remains almost unchanged under the identical condition for 2 weeks. The rejections are above 99% for CR and above 91% for RB. The above tests confirm that the MOPM- Fe^{3+} /PAN membrane can withstand the existence of salts in solution and shows excellent long-term operational stability benefited from the strong P-O-Fe coordination bond.

Figure R5. (a) and (b) SEM images of MOPM-Fe³⁺/PAN after immersed in salt solution for 2 weeks. Scale bar: 500 nm. (c) and (d) Long-term stability of MOPM-Fe³⁺/PAN membrane in in salt solution. (1000 ppm Na₂SO₄/1000 ppm NaCl).

The abovementioned experimental results have been added in **manuscript** and **Supplementary Material** as follows.

Besides, the strong Fe-O-P binding also brings about structural stability under saline conditions (Supplementary Figure 25), low metal leakage during filtration (Supplementary Table 6) and sufficient mechanical strength (Supplementary Figure 26). The comprehensive stability in various harsh conditions makes the MOPM-Fe³⁺/PAN membrane a promising candidate for practical water purification.

Supplementary Figure 25. (a) and (b) SEM images of MOPM-Fe³⁺/PAN membrane after immersed in salt solution for 2 weeks. Scale bar: 500 nm. (c) and (d) Long-term stability of MOPM-Fe³⁺/PAN membrane in Na₂SO₄ and NaCl solution.

4. What would happen to the membranes if the wastewater has high PH values? For example, if NaOH is presented, would it neutralize the phytic acid and change the nanostructure of the membranes?

Reply:

This is indeed an important issue. As a matter of fact, we have described this issue in **Figure 23** of the original **Supplementary Material (Figure S24 in the revised Supplementary Material)**. Under alkali condition, the existence of OH⁻ will promote the deprotonation of phosphate groups, which is favorable for their coordination binding with Fe³⁺ ions. Therefore, the metal-organophosphate

complexes possess higher stability constant (K) and become more stable under alkali condition: pH=5.0, $K=1.60\times 10^{18}$; pH=9.0, $K=1.58\times 10^{19}$ (*J. Inorg. Biochem.*, 2005, 99(3): 828-840).

Supplementary Figure 24. SEM images of MOPM-Fe³⁺/PAN membrane after immersed in varied pH solutions for 24 h. The PA concentration, PA/Fe ratio and assembly time were 0.015 mg/mL, 1:7 and 60 min, respectively. Scale bar: 500 nm.

Under alkali condition (pH=10.0 and 12.0), defect-free, rougher surface structures could be observed, which is probably because the deprotonation of phosphate groups promotes the further coordination with Fe³⁺ ions and increases the compactness of metal-organic networks (*Science*, 2013, 341(6142): 154-157). In brief, the existence of NaOH does not deteriorate the filtration performance of MOPM-Fe³⁺/PAN membrane owing to the defect-free structure as shown in Figure 4 (b) of the revised **manuscript** and original **manuscript**.

Fig. 4 (b) Stability performance of MOPM-Fe³⁺/PAN membrane under varied pH conditions by immersing membrane in HCl and NaOH solution for 24 h.

Reviewer #2 (Remarks to the Author):

This manuscript reports metal organophosphate membranes (MOPMs) fabricated by mixing natural phytic acid (PA) with heavy metal ions (including Ag⁺, Zn²⁺, Ni²⁺, Fe³⁺, and Zr⁴⁺) for nanofiltration. Five MOPM-metal ion membranes were prepared on PAN substrate, it seems that only MOPM-Fe³⁺ membrane could have a thickness down to sub-10 nanometers, and high water permeance up to 109.8 L m⁻² h⁻¹ bar⁻¹, high rejection (>95 %) of dyes with molecular sizes > 2 nm. The authors also claim that MOPM-Fe³⁺ membranes have very good stability and regenerability because of the stronger metal-organic coordination between metal ions and organophosphate ligands. Therefore, I believe that this manuscript would be potentially accepted in Nature Communications after some revisions.

Reply:

Thank the reviewer for the highly positive comments.

1. The first major comment is the “sub-10 nm” in the manuscript title, which is kind of confusing. It’s difficult to tell whether the “sub-10 nm” refers to the pore size or the thickness of the membrane without any context.

Reply:

Thanks for the reviewer’s comment. We have reviewed relative literatures in which “sub-10 nm” is used in the titles, and found that in the field of membrane separation, the phrase “sub-10 nm” always refers to the thickness of membrane and has been commonly used in titles of literatures to describe ultrathin membranes, e.g. “**Sub-10 nm** polyamide nanofilms with ultrafast solvent transport for molecular separation (*Science*, 2015, 6241:1347-1351)”, “Large-area, transferable **sub-10 nm** polymer membranes at the air-water interface (*Nano Res.*, 2018, 11: 3833-3843)” and “Synthesis of **sub-10 nm** two-dimensional covalent organic thin film with sharp molecular sieving nanofiltration” (*ACS Appl. Mater. Inter.*, 2018, 10:12295-12299), etc. Besides, since the pore sizes of membranes for molecular separation such as tight ultrafiltration membrane, nanofiltration membrane, reverse osmosis membrane, gas separation membrane and pervaporation membrane are all much smaller than 10 nm, researchers in these membrane fields do not need to emphasize “sub-10 nm” in

describing the pore size. Therefore, based on the literature review and the consensus in the membrane field, “sub-10 nm” could be used and will not cause the confusion for the readers.

2. The second major comment is about the membrane fabrication. The authors claim that the self-assembled molecular layer is formed by H-bond interaction between the carboxyl groups on hydrolyzed PAN and the phosphate groups on PA. Does the pH value of the PA solution effect on the membrane quality and thickness. As we know, if the solution pH value is higher than the isoelectric points of the PAN and PA, both the carboxyl groups and phosphate groups are negatively charged, and there should be electrostatic repulsion between them, which is adverse to the formation of PA layer on the substrate. In addition, the metal ions were added subsequently into the PA solution, it is possible that the metal ions would coordinate with carboxyl groups on PAN firstly, serving as cross-linkers, and then coordinate with phosphate groups via the unsaturated metal sites.

Reply:

This is a very important question. To clarify the assembly process of MOPMs, we measured the pH value of assembly solution and investigated the interaction between the substrate and PA molecules. Besides, we confirmed the existence of PA molecules on the PAN substrate by elemental mapping and further coordination assembly.

Since the natural phytic acid possesses plenty of protons, the pH value of the assembly solution is around 2.20 (PA content 0.015 mg/mL). This condition allows $\text{FeCl}_3 \cdot 6\text{H}_2\text{O}$ to fully dissociate into Fe^{3+} ions for coordination (**Figure R6**). Under higher pH value, the ferric hydroxides are not suitable for coordination and unable to assemble high-quality membranes.

Figure R6. The existence forms of Fe^{3+} ion under different pH conditions.

As shown in **Figure R7**, we measured the isoelectric point (IP) of hydrolyzed PAN substrate (IEP=2.6) and deduced that the PAN substrate is positively charged in assembly solution (zeta potential \approx 6.70 mV). The deprotonated PA molecules are negatively charged. Therefore, there is no electrostatic repulsion to hamper the self-assembly of PA layer on hydrolyzed PAN substrate. The absorbed PA molecules on PAN substrate were verified by P elemental mapping, as shown in **Figure**

R8.

Figure R7. Zeta potential of PAN substrate under varied pH condition.

Figure R8. The P element distribution on PA-treated PAN substrate.

Additionally, after rinsed with DI water for 10 min, the PA-treated PAN was immersed into the Fe^{3+} solution (4.2 mg/ml, 30 mL) for coordination (60 min). As shown in **Figure R9**, this process can still obtain MOPM- Fe^{3+} on PAN substrate. However, due to the absence of ligands in assembly solution, obvious defects can be observed on membrane surface. The above result further verifies the existence the absorbed PA on the PAN substrate.

Figure R9. SEM images of (a) PA-treated PAN substrate and (b) MOPM-Fe³⁺/PAN membrane prepared by PA-treated PAN substrate. Scale bar: 500 nm.

Considering that the hydrolyzed PAN substrate is positively charged in assembly solution, we deduced that the abundant H⁺ in solution hinders the deprotonation of carboxyl groups and thus the coordination with positively-charged Fe³⁺ ions. Besides, the organophosphate groups attract more Fe³⁺ ions than carboxyl groups because of their strong interaction with Fe³⁺ ions. After washed with DI water, the weakly bonded Fe³⁺ ions are most probably removed. And the chance of the coordination between Fe³⁺ and deprotonated carboxyl groups on PAN is quite limited.

The relevant information has been added in the **Supplementary Material** as follows.

Supplementary Figure 1. (a) The surface morphology of PAN substrate. Scale bar: 500 nm. Inset: Pore size distribution of PAN substrate, estimated by an image analysis software titled Nano Measure. (b) The zeta potential of PAN substrate under varied pH condition. (c) The cross-section TEM image of PAN substrate. Scale bar: 20 nm. (d) The cross-section SEM image of PAN substrate. Scale bar: 500 nm (e) SEM image of PA-treated PAN substrate. (f) EDS result of PA-treated PAN membrane. Inset: Distribution of phosphorus on PA-treated PAN substrate.

Supplementary Table 7. Summary of PA concentration (mg/mL), metal salt content (mg/mL),

assembly time (min) and pH value of assembly solution for fabricating MOPMs. The solute content during assembly was calculated based on solution volume of 30 mL.

Membrane name	PA concentration (mg/mL)	Metal salt type	Metal salt content (mg/mL)	Mole ratio (PA:M)	Assembly time (min)	pH value
MOPM-Ag ⁺	0.015	AgNO ₃	2.8	1:7	60	2.22
MOPM-Zn ²⁺	0.015	ZnCl ₂	2.3	1:7	60	2.20
MOPM-Ni ²⁺	0.015	NiCl ₂ ·6H ₂ O	3.9	1:7	60	2.24
MOPM-Fe ³⁺	0.015	FeCl ₃ ·6H ₂ O	4.2	1:7	60	2.21
MOPM-Zr ⁴⁺	0.015	Zr(NO ₃) ₄ ·5H ₂ O	7.0	1:7	60	2.23
Condition experiment of MOPM-Fe³⁺						
MOPM-Fe ³⁺ -1	0.015	FeCl ₃ ·6H ₂ O	0.3	1:0.5	60	2.23
MOPM-Fe ³⁺ -2	0.015	FeCl ₃ ·6H ₂ O	0.6	1:1	60	2.21
MOPM-Fe ³⁺ -3	0.015	FeCl ₃ ·6H ₂ O	1.2	1:2	60	2.24
MOPM-Fe ³⁺ -4	0.015	FeCl ₃ ·6H ₂ O	3.0	1:5	60	2.22
MOPM-Fe ³⁺ -5	0.015	FeCl ₃ ·6H ₂ O	4.2	1:7	60	2.20
MOPM-Fe ³⁺ -6	0.030	FeCl ₃ ·6H ₂ O	8.4	1:7	60	2.14
MOPM-Fe ³⁺ -7	0.045	FeCl ₃ ·6H ₂ O	12.6	1:7	60	2.19
MOPM-Fe ³⁺ -8	0.015	FeCl ₃ ·6H ₂ O	6.3	1:10	60	2.24
MOPM-Fe ³⁺ -9	0.015	FeCl ₃ ·6H ₂ O	4.2	1:7	40	2.25
MOPM-Fe ³⁺ -10	0.015	FeCl ₃ ·6H ₂ O	4.2	1:7	50	2.22
MOPM-Fe ³⁺ -11	0.015	FeCl ₃ ·6H ₂ O	4.2	1:7	70	2.21
MOPM-Fe ³⁺ -12	0.015	FeCl ₃ ·6H ₂ O	4.2	1:7	80	2.20

3. The third major comment is about the fabrication process of using heavy metal ions to joint PAs, which was carried out at 60 oC. Is the temperature essential for the final quality of the MOPM membranes? It seems that 60 oC is only suitable for fabricating high-quality the MOPM-Fe³⁺/Zr⁴⁺ membranes as shown in Figure 1f. Therefore, more details should be given in the supporting why the

authors cannot get defect-free membranes based on other three metal ions under this condition.

Reply:

Upon the reviewer's request, we prepared MOPM-Ag⁺, MOPM-Zn²⁺ and MOPM-Ni²⁺ under higher thermal curing temperature (90 °C). As shown in **Figure R10**, the defects in these membranes can be still observed. This hints that the structural defects are determined by their chemical nature rather than treatment temperature. Because the metal ions with low ionization potential (Ag⁺, Zn²⁺ and Ni²⁺) possess weaker coordination ability with ligands, the highly coordinated network is difficult to be generated. As shown in **Supplementary Table 3**, at the same metal ion and ligand concentration, the MOPM-Ag⁺, MOPM-Zn²⁺ and MOPM-Ni²⁺ exhibit lower metal ion ratio than MOPM-Fe³⁺ and MOPM-Zr⁴⁺, which account for the inferior coordination ability of Ag⁺, Zn²⁺ and Ni²⁺ ions.

Figure R10. SEM images of MOPMs prepared under high curing temperatures (90 °C). Scale bar: 500 nm.

Supplementary Table 3. Summary of elemental ratio of MOPMs in this work.

Membrane name	Metal salt type	Mole ratio ^a (PA:M)	Mole ratio ^b (PA:M)
MOPM-Ag ⁺	AgNO ₃	1:7	1:1.1
MOPM-Zn ²⁺	ZnCl ₂	1:7	1:1.5
MOPM-Ni ²⁺	NiCl ₂ ·6H ₂ O	1:7	1:1.4

MOPM-Fe ³⁺	FeCl ₃ ·6H ₂ O	1:7	1:3.3
MOPM-Zr ⁴⁺	Zr(NO ₃) ₄ ·5H ₂ O	1:7	1:4.1

^a The PA:M mole ratio in assembly solution.

^b The mole ratio of MOPMs detected by EDX, where M represented Ag, Zn, Ni, Fe and Zr element.”

Supplementary Figure 5. SEM images of MOPMs prepared under high curing temperatures (90 °C).

Scale bar: 500 nm.

4. The fourth major comment is about the dyes used for rejection testing and the mechanism for the high dye rejection. The selected dyes for rejection testing are methyl blue (1.6×2.0 nm), congo red (1.1×2.2 nm), and alcian blue (1.4×2.2 nm). They have very close molecular sizes, but the MOPM-Fe³⁺ membrane showed the highest rejection for congo red as shown in Figure 4. Why the authors chose these three dyes? How about the membrane performance for filtration much smaller organic molecules?

As shown in Figure S 14. There are sulfonate groups on methyl blue and congo red, and positive charges on alcian blue. As a result, the negatively charged MOPM-Fe³⁺/PAN membranes may reject the negatively charged methyl blue and congo red molecules by electrostatic repulsion, while adsorb positively charged alcian blue molecules through electrostatic attraction onto the negatively membrane surface or into the pores of the membranes, showing the lowest rejection of alcian blue. Therefore, a size sieving effect is not suitable to explain the rejection mechanism of the MOPM

membranes. Some experimental evidence should be provided to confirm the adsorption and repulsion effects can be ignored if the authors think the size sieving effect is the main reason for dye rejection.

Reply:

Based on the reviewer's valuable guidance, we supplemented another two kinds of organic dyes featuring smaller size to conduct the filtration experiment (**Figure R11**).

Figure R11. Structural information of organic dyes used in this work. Methyl blue (MB, 1.62×2.03 nm), Congo red (CR, 1.10×2.20 nm), Alcian blue (AB, 1.42×2.20 nm), Rose Bengal (RB, 1.06×1.08

nm) and Orange GII (OG, 0.74×1.07 nm).

These dyes could be classified based on size: small-size dye (<1 nm, OG), middle-size dye (~1 nm, RB) and large-size dye (>2 nm, MB/CR/AB). Then we evaluated the membrane performance for filtrating OG and RB. The optimized MOPM-Fe³⁺/PAN membrane exhibits the following dye rejections: MB (95.9%), CR (100%), AB (96.9%), RB (92.5%) and OG (88.5%). Besides, the dye adsorption was also evaluated to elucidate the separation mechanism, as shown in **Figure R12**. The five kinds of organic dyes feature similar adsorption amount, and the positively-charged AB exhibits the highest adsorption amount on the negatively-charged MOPM-Fe³⁺/PAN membrane. The size-sieving effect and electrostatic interaction are the major factors affecting the rejection. Considering that the dye rejection follows the order of small-size dye (<1 nm, OG) < middle-size dye (~1 nm, RB) < large-size dye (>2 nm, MB/CR/AB), the size of organic dye still plays a dominant role during filtration. Additionally, by comparing the rejection of MB and CR with similar size, we found that the CR with higher negative charge had higher rejection, indicating that the electrostatic repulsion also contributed to the high dye rejection.

Figure R12. The dye adsorption of MOPM-Fe³⁺/PAN membrane.

The **manuscript** and **Supplementary Material** have been revised as follows.

Permselectivity and stability of MOPM/PAN membranes. The permselective performance of resultant MOPM/PAN membranes was evaluated in a filtration apparatus (Supplementary Figure 14) using five kinds of typical organic dyes with varied molecular size (0.74–2.20 nm) to simulate organic wastewater (Supplementary Figure 15). The pristine PAN substrate and PA-treated PAN substrate showed low dye rejections (30–60%) owing to their large pores (Supplementary Figure 16). As shown in Fig. 3a, the MOPM/PAN membranes employing Ag^+ , Zn^{2+} and Ni^{2+} as coordination metal ions still show inferior dye rejections similar to PAN substrate due to the structural defects (Fig. 1e). Considerably higher dye rejections (88.5–100%) are achieved by using the metal ions (Fe^{3+} and Zr^{4+}) with larger ionization potential which led to defect-free membrane structure. The permselective behavior of MOPM- Fe^{3+} /PAN membrane with potentially higher water permeance than the less hydrophilic MOPM- Zr^{4+} /PAN was further investigated.

The solute rejections of MOPM- Zr^{4+} /PAN are in an order of small-size dye (<1 nm, OG) <middle-size dye (~1 nm, RB) <large-size dye (>2 nm, MB/CR/AB), indicating the size exclusion plays an important role during filtration. However, the influence of electrostatic interaction is a nonnegligible factor for the charged solute rejection of the negatively-charged MOPM- Fe^{3+} /PAN besides the size-based selectivity³¹. Specifically, even though CR, MB and AB have similar size (1.10–2.20 nm), the rejections follow the order of CR>MB>AB due to their different charge property. The AB with positive charge prompts to be attracted by the negatively-charged MOPM- Fe^{3+} skin layer and thus resulting in a higher solute permeance and hence lower rejection. As for the negatively-charged CR and MB, the membrane shows a higher rejection for CR with a more negative charge of -40.2 ± 0.7 mV than MB with a relatively less negative charge of -12.0 ± 0.6 mV. This indicates that the electrostatic repulsion contributes to high dye rejection. Moreover, the increased Fe^{3+} content in assembly solution enhances the CR rejection of MOPM- Fe^{3+} /PAN gradually from 94.1% to 100% (Fig. 3b) along with the decreased permeance due to the denser membrane structure (inset in Fig. 2d). Fig. 3c shows that the ultrathin and dense MOPM- Fe^{3+} /PAN (PA/Fe ratio=1:7) with high water permeance of $\sim 109.8 \text{ L m}^{-2} \text{ h}^{-1} \text{ bar}^{-1}$ could completely reject CR in feed (100 ppm) and yield purified water (~ 0 ppm) and concentrated retentate. As shown in Supplementary Figure 17 and 18, the membrane also exhibits high rejections for AB (95.9%) and MB (96.9%) as well as moderate rejection for RB (92.5%) and OG

(88.5%).

Fig. 3 Permselectivity of MOPM/PAN membranes. **a,b** Filtration performance of MOPM/PAN membranes with

different coordinated metal ions (**a**) and MOPM-Fe³⁺/PAN membranes with varied PA/Fe ratio (**b**). 100 ppm of

Methyl blue (MB, 1.62×2.03 nm), Congo red (CR, 1.10×2.20 nm), Alcian blue (AB, 1.42×2.20 nm), Rose Bengal (RB,

1.06×1.08 nm) and Orange GII (OG, 0.74×1.07 nm) solution as feed. **c** Ultraviolet-visible spectra of Congo red in

feed, retentate and filtrate. PA/Fe ratio=1:7. Inset in **c**: Digital photo images of feed (Fe), retentate (Re) and filtrate

(Fi) (top left) and molecular structure of Congo red (bottom right). **d** Ultraviolet-visible absorption spectra of

graphene oxide quantum dots (GQDs) in feed, retentate and filtrate. PA/Fe ratio=1:0.5. Inset in **d**: Size distribution

of GQDs (top left), digital photo images of feed (Fe) and filtrate (Fi) (top right) and TEM image of GQDs (bottom).

Scale bar: 10 nm. **e** Water permeance of MOPM-Fe³⁺/PAN membranes with different layer number of MOPM-Fe³⁺.

PA/Fe ratio=1:7. Red line was the best exponential fit. Inset in **e**: Mass transport resistance as a function of

thickness for MOPM-Fe³⁺/PAN membranes with different skin layer thicknesses. Red line was the best linear fit. **f**

Filtration performance of state-of-the-art polymeric membranes for water purification in literatures.

Supplementary Figure 18. Ultraviolet-visible absorption spectra of (a) Methyl blue (MB), (b) Alcian blue (AB), (c) Rose Bengal (RB), (d) Orange GII (OG) and (e) GQDs in feed and filtrate of MOPM-Fe³⁺/PAN membrane. Inset: Digital photo images of feed (Fe) and filtrate (Fi) (top left) and molecular structure of MB and AB. The filtrate of cycle 1 (5 mL feed) was used as the feed of cycle 2. (f) The dye adsorption of MOPM-Fe³⁺/PAN membrane. The PA concentration, PA/Fe ratio and assembly time were fixed at 0.015 mg/mL, 1:7 and 60 min, respectively.

5. Even though the filtration performance of MOPM-Fe³⁺/PAN membranes is remarkable, the performance of pristine PAN and PA treated PAN should be included as comparative experiments.

Reply:

Upon the reviewer's request, we tested the filtration performance of the PAN substrate and PA-treated PAN substrate as shown in **Supplementary Figure 16 (a)** of the revised **Supplementary Material**. The PAN and PA-treated PAN exhibited water permeance of $\sim 300 \text{ L m}^{-2} \text{ h}^{-1} \text{ bar}^{-1}$ and low dye rejections of around 30-60%.

Supplementary Figure 16. (a) Filtration performance of PAN and PA-treated PAN. (b) and (c) Filtration performance of MOPM-Fe³⁺/PAN membranes with varied PA concentration and assembly time, respectively. The dye concentration was 100 ppm. The PA concentration, PA/Fe ratio and assembly time were fixed at 0.015 mg/mL, 1:7 and 60 min, respectively.

6. For practical membrane-based filtration, the high water permeance, long-term stability and anti-fouling ability really matter. Meanwhile, the membrane mechanical strength should be

considered.

Reply:

Upon the reviewer's request, the Young's modulus (E) of MOPM-Fe³⁺/PAN membrane was measured to evaluate the mechanical strength of skin layer by Atomic Force Microscope according to the method reported in literature (*ACS Appl. Mater. Inter.*, 2017, 9: 2966-2972). The MOPM-Fe³⁺ exhibits a root-mean-square Young's modulus of 8.319 GPa. Besides, the underlying PAN substrate and robust nonwoven fiber can ensure the overall mechanical stability of thin film composite membrane. The results have been provided in **Supplementary Material**.

Supplementary Figure 26. Young's modulus (E) of MOPM-Fe³⁺/PAN membrane. The PA concentration, PA/Fe ratio and assembly time were 0.015 mg/mL, 1:7 and 60 min, respectively.

7. Fig S2 in the supporting information cannot be seen clearly. High quality image should be provided.

Reply:

Upon the reviewer's request, the high-resolution images were provided in the revised **Supplementary Material** as follows.

Supplementary Figure 2. (a) Simulated coordination mode between metal ion and methyl phosphate. (b) EDX mapping and (c) related SEM images of MOPMs. Scale bar: 5 μm .

8. Since the authors used the MOPMs fabricated based on heavy metal ions for water treatment, it would be better to provide some data of the amount of heavy metal ions in the permeate water.

Reply:

Upon the reviewer's request, we measured the metal heavy leakage (Ag^+ , Zn^{2+} , Ni^{2+} , Fe^{3+} and Zr^{4+})

of MOPM/PAN membranes by analyzing the metal element content of 30 mL permeate water with inductively coupled plasma (ICP), as shown in **Table R1**. The MOPM-Fe³⁺/PAN with the highest filtration performance exhibited only 0.0047 ppm metal leakage in permeate water, lower than the limit of drinking water standard in China, World Health Organization (WHO) and USA.

Table R1. Metal ion leakage during filtration (30 mL filtrated DI water).

Membrane type	Detected metal ion concentration	Limiting metal ion concentration
MOPM-Fe ³⁺	0.0047 ppm, Fe ³⁺	0.3 ppm ^a ; 0.3 ppm ^b ; 0.3 ppm ^c
MOPM-Ag ⁺	0.0313 ppm, Ag ⁺	0.05 ppm ^a ; 0.1 ppm ^b ; 0.1 ppm ^c
MOPM-Zn ²⁺	not detected, Zn ²⁺	1.0 ppm ^a ; 3.0 ppm ^b ; 5.0 ppm ^c
MOPM-Ni ²⁺	not detected, Ni ²⁺	0.02 ppm ^a ; 0.02 ppm ^b
MOPM-Zr ⁴⁺	not detected, Zr ⁴⁺	N/A

^a Drinking-water quality standard (GB5749-2006), (2007, China).

^b Guidelines for drinking-water quality 4thed, (2011, World Health Organization, WHO).

^c National Secondary Drinking Water Regulations, (2007, USA).

The relevant information as been added into the revised **manuscript** and **Supplementary Material** as follows.

Besides, the strong Fe-O-P binding also brings about structural stability under saline conditions (Supplementary Figure 25), low metal leakage during filtration (Supplementary Table 6) and sufficient mechanical strength (Supplementary Figure 26). The comprehensive stability in various harsh conditions makes MOPM-Fe³⁺/PAN membrane a promising candidate for practical water purification.

The metal ion leakage of the five types of as-prepared MOPM/PAN membranes was measured by detecting metal element (Ag⁺, Zn²⁺, Ni²⁺, Fe³⁺ and Zr⁴⁺) in filtrated water. 30 mL DI water was filtrated through MOPM/PAN membranes and collected for metal element content (ppm) measurement by inductively coupled plasma (ICP, Leeman Prodigy, USA).

Supplementary Table 6. Metal ion leakage during filtration (30 mL filtrated DI water).

Membrane	Detected metal ion	Limiting metal ion concentration
----------	--------------------	----------------------------------

type	concentration	
MOPM-Fe ³⁺	0.0047 ppm, Fe ³⁺	0.3 ppm ^a ; 0.3 ppm ^b ; 0.3 ppm ^c
MOPM-Ag ⁺	0.0313 ppm, Ag ⁺	0.05 ppm ^a ; 0.1 ppm ^b ; 0.1 ppm ^c
MOPM-Zn ²⁺	not detected, Zn ²⁺	1.0 ppm ^a ; 3.0 ppm ^b ; 5.0 ppm ^c
MOPM-Ni ²⁺	not detected, Ni ²⁺	0.02 ppm ^a ; 0.02 ppm ^b
MOPM-Zr ⁴⁺	not detected, Zr ⁴⁺	N/A

^a Drinking-water quality standard (GB5749-2006), (2007, China).

^b Guidelines for drinking-water quality 4thed, (2011, World Health Organization, WHO).

^c National Secondary Drinking Water Regulations, (2007, USA).

REVIEWERS' COMMENTS:

Reviewer #1 (Remarks to the Author):

The authors have satisfactorily addressed my prior comments.

Reviewer #2 (Remarks to the Author):

I appreciate all the efforts that the authors have made to address the concerns on the work. I believe the explanations based on the additional experimental results can well solve my major concerns about the membrane fabrication process and dyes rejection mechanisms. Therefore, I suppose the manuscript is now suitable for publication.

Manuscript ID: NCOMMS-19-15356A

TITLE: Metal-coordinated sub-10 nm membranes for water purification

Response to the reviewers' comments

Reviewer #1 (Remarks to the Author):

The authors have satisfactorily addressed my prior comments.

Reply:

Thank the reviewer for the highly positive comments and the efforts in reviewing this manuscript.

Reviewer #2 (Remarks to the Author):

I appreciate all the efforts that the authors have made to address the concerns on the work. I believe the explanations based on the additional experimental results can well solve my major concerns about the membrane fabrication process and dyes rejection mechanisms. Therefore, I suppose the manuscript is now suitable for publication.

Reply:

Thank the reviewer for the highly positive comments and the efforts in reviewing this manuscript.